# XIMAGENET-12: AN EXPLAINABLE AI BENCHMARK DATASET FOR MODEL ROBUSTNESS EVALUATION

## ABSTRACT

The lack of standardized robustness metrics and the widespread reliance on numerous unrelated benchmark datasets for testing have created a gap between academically validated robust models and their often problematic practical adoption. To address this, we introduce XIMAGENET-12, an explainable benchmark dataset with over 200K images and 15,600 manual semantic annotations. Covering 12 categories from ImageNet to represent objects commonly encountered in practical life and simulating six diverse scenarios, including overexposure, blurring, color changing, etc., we further propose a novel robustness criterion that extends beyond model generation ability assessment. This benchmark dataset, along with related code, is available at `https://sites.google.com/view/ximagenet-12/home`. Researchers and practitioners can leverage this resource to evaluate the robustness of their visual models under challenging conditions and ultimately benefit the demands of practical computer vision systems.

## 1 INTRODUCTION

Visual models have widely been utilized in the industrial sector, especially OEMs (Original Equipment Manufacturers) specializing in large-scale computer vision applications for defect detection(10), image enhancement (44), quality control(24), and predictive maintenance topics (41)(20). Those models often touted as highly effective in academic settings that excel on large benchmark datasets, tend to exhibit surprising shortcomings when deployed in real-world scenarios (20; 40). Issues such as sensitivity to changes in lighting conditions, background interference, shifts in object positions, and the presence of previously unseen noise or artificial camera disturbances in the background are common challenges that supposedly stable models, as claimed in academic papers, often fail to address adequately(41)(24).

Researchers from academia often lack a readily available and interpretable dataset to explore the specifics of how AI utilizes information for content understanding and which cues it is really sensitive to. Creating such a dataset typically requires extensive manual labeling efforts and resources. Therefore, we decided to use the well-known ImageNet dataset (6) as a foundation for our work, as it serves as a cornerstone in the field of convolutional neural networks (CNNs) and is extensively used to evaluate visual models. However, ImageNet (6) is primarily a classification dataset, meaning it provides only category labels for each class and sometimes includes some images that do not belong to their designated categories. Recent advancements in Explainable machine learning (30) have led to various improvements in this direction, such as the IOG dataset (42), which starts with ImageNet and filters 1,000 images per class using neural networks to create virtual labels. There's also the ImageNet-9 dataset (35), which selects nine classes and explores the impact of backgrounds on foreground objects, using neural networks to assign labels for each class and experimenting with background replacements. Some approaches (14) even utilize graph neural networks (GNNs) to generate semantic labels for ImageNet. While these efforts have been inspiring, they still suffer from significant drawbacks. Using neural networks to generate pseudo labels (18) (38) (4) can

introduce significant inaccuracies (see Fig. 6. Furthermore, datasets like ImageNet-9 (35) do not deeply investigate how backgrounds influence model performance; they merely suggest that backgrounds are not always noise and can sometimes be helpful, touching upon the finding that more robust models tend to rely less on backgrounds without quantitative comparison.

Hence, our team embarked long journey involving over 20 contributors to the annotation process and designed our dataset to cover 12 major categories, including objects that fly, swim, and are commonly found in everyday life, as well as some intricate multi-background objects. In total, we meticulously annotated over 12,000 images at the pixel level and conducted further research into scenarios like background replacement, complete background removal for enhanced validation, background blurring, foreground blurring, and color changes to simulate object shifts caused by camera vibrations in industrial production processes. Additionally, we explored artificially rendered backgrounds and backgrounds generated by the Stable Diffusion model (28) to examine the model's sensitivity to different image variations.

The creative highlight of our work lies in the in-depth exploration of the state-of-the-art (SOTA) models, including Transformer-based models and segmentation models as well to analyze their sensitivity to image backgrounds. We have delved into defining model robustness, as existing robustness tests (23), (39) typically involve stacking massive amounts of irrelevant data that often differ significantly from the objects' backgrounds encountered in industrial settings. Strong performance on benchmark datasets does not necessarily translate to high accuracy in real-world industrial projects (34). Furthermore, model robustness should encompass the ability to recognize foreground objects accurately even in the presence of various new variables introduced by background interference. Therefore, our work combines dataset creation, quantitative model analysis, and the innovative proposal of a robustness comparison formula, allowing for a comparative evaluation of model robustness and providing guidance for practical engineering model usage.

Our main contributions can be summarized as follows:

- We annotate and GenAI generate a dataset for the explainable AI domain and Gen-AI domain, which possesses various annotation categories, relatively high image information quality, and public availability.
- We propose the model robustness score formal schema based our dataset for a better selection of models for industry applications.
- We quantitatively analyses and benchmark SOTA object detection, classification, and segmentation models using our dataset encompasses both CNN- and Transformer-based architectures, aimed at validating the models' robustness while prioritizing explainable AI principles.

## 2 DATASET CREATION PROCEDURE:

In this section, we will introduce our dataset from three aspects: (1) the process of image capturing, (2) the overall properties of our dataset, and (3) the composition of our dataset.

### 2.1 DATA COLLECTION

12 categories are selected from the ImageNet dataset as the fundamental image, which represents objects frequently encountered in everyday life. Then we synthesize 6 scenarios (refer to Fig.1) to comprehensively evaluate model robustness. These scenarios include blurred images, colored images, segmented images, transparent images, images with randomly generated backgrounds, and images with AI-generated backgrounds.

**Colored images:** In this scenario, we enrich the diversity of dataset by generating images from grayscale, single-channel (R, G, B), rainbow images, and different brightness of both background and foreground. The

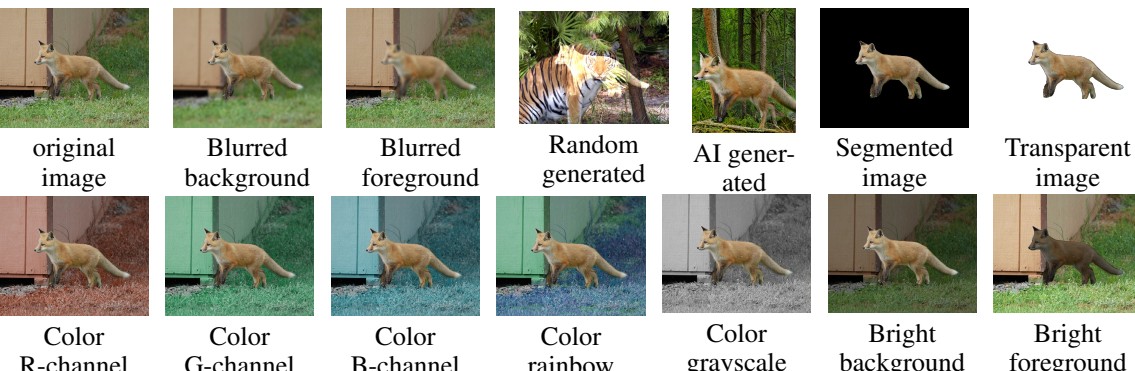

Figure 1: XIMAGENET-12 sample for 6 scenarios: Blur, Random generated background, AI-generated background, Segmentated, Transparent and Color images. Over 200K Images in total.

process of grayscale and single-channel is to keep the object in a color image unchanged while altering the background to grayscale or individual RGB channels. The function **addWeighted** in OpenCV is used to apply Unsharp Masking (USM) to enhance brightness and sharpness. The rainbow image is generated by converting the image to HSV color space and changing the hue for the background. Color images can simulate the changes in lighting and color conditions in the real world. Keeping the object of interest unchanged allows for better analysis of the object's characteristics and reduces distractions from the background.

**Blurred images:** By combining with mask information, we propose two ways of blurring images: blurring the image background and blurring the foreground object. The function **GaussianBlur** in OpenCV is used to apply the blur properties to the image. In real applications, blur often happens when a camera is small shifted, which leads to a loss of fine-grained details and compressed information. By weakening the background/foreground we have a preliminary understanding of the impact of background and foreground on model inference. (7).

**Segmented images:** This scenario is composed of segmented objects only. To be specific, we create the new image with RGBA 4 channels, and the RGB channel pixel information will be unchanged for the foreground, and the alpha channel based on the binary mask. For example, if the pixel at (x, y) in the original image is (r, g, b), based on the binary mask, it will become (r, g, b, 255), if it is in the background then (0, 0, 0, 255). This scenario can enhance the interpretability of various computer vision tasks by highlighting objects of interest and separating them from the background (5).

**Transparent images:** There is a serious problem when using segmented images to generate AI background images, the **Stable Diffusion XL model** (26) will also take the alpha channel value into account (28), leading to lots of black spots surrounding the object. In order to overcome the problem, we create transparent images. Similar to the segmented scenario, we also create a new image with RGBA 4 channels, based on the binary mask, the background will be completely removed and the object will be kept the same. For example, if the pixel at (x, y) in the original image is (r,g,b), based on the binary mask, it will become (0, 0, 0, 0) if it is in the background, otherwise, it will be (r, g, b, 255).

**Images with randomly generated backgrounds:** In this scenario, images are generated with the photographer's real image as a background and blended with the object from the original image. Specifically, When selecting background pictures, we deliberately choose pictures that are closely related to the natural environment. This scenario simulated the challenge of distorting the real environment. Because of **addWeighted** blending, **resize**, **random Shuffle and position** function, we can give different weights to different images, which can make the image transparent or translucent according to the weight added. As a result, the

model becomes more robust and likely to generalize better to real-world scenarios where the background is unpredictable.

**Images with AI-generated backgrounds:** Except randomly generating unrelated backgrounds, we also create a scenario where image backgrounds are generated by AI. Specifically, the backgrounds are created using **Playground AI** (25) by providing transparent images as input, following the Stable Diffusion XL model (26) for text-to-image generation with appropriate prompts. Different prompts are applied to different classes (See Fig.7), some very useful tips: Using keywords such as 'National Geographic Magazine' or 'National Oceanic Magazine' can increase the authenticity of the generated background; Adding specific and appropriate environmental information to the prompt can make the generated background and objects better integrated. By using a diffusion model and introducing unexpected or extreme background variations, we can assess whether the model is resilient against potential adversarial attacks involving background manipulation.

## 2.2 DATASET PROPERTIES

Aiming to show the properties of XIMAGENET-12 clearly, we illustrate the causality in the following perspectives:

**Practicality**: The 12 categories selected from ImageNet represent objects commonly encountered in practical life, and the scenarios we created accurately simulate problems that may be encountered in industry, such as brightness change, and background interference.

**Scenarios**: In addition to the fundamental images, XIMAGENET-12 is composed of 6 scenarios, which are colored images (gray, single-channel, rainbow, brightness change), blurred images (blurred background, blurred foreground), images with randomly generated backgrounds, images with AI-generated backgrounds, segmented images, and transparent images.

**Foreground invariance**: This invariance is relative to the variability of the background, through the whole XIMAGENET-12, the background is changed in each scenario, while most of the foreground object remains the same.

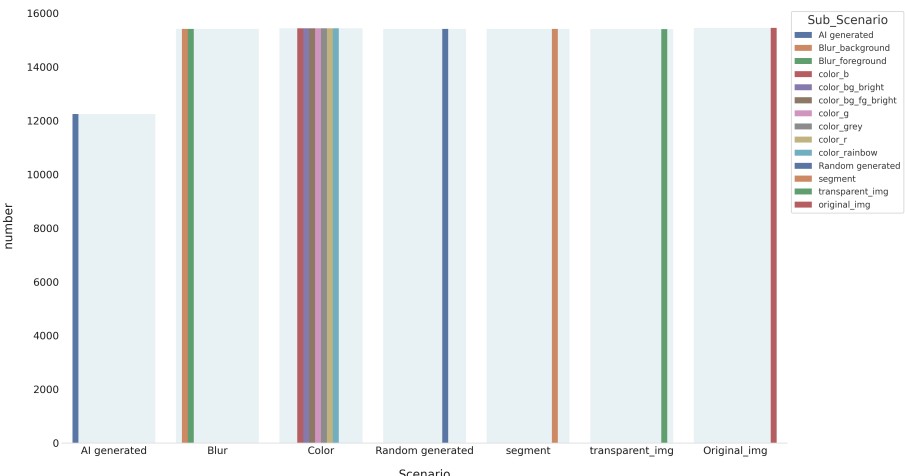

Figure 2: The scenario distribution of XIMAGENET-12

## 2.3 Dataset Distribution

Fig.2 illustrates the distribution of our dataset across various scenarios, including fundamental images. We've curated 12 classes from ImageNet, each containing around 1,300 samples, resulting in a total selection of 15,450 images from ImageNet. As images from six scenarios are derived from fundamental images, each transformation within a scenario results in approximately 15,400 images. However, generating AI background images presented challenges, such as dealing with scenarios featuring multiple objects in a single image and the presence of small or incomplete objects. Nevertheless, through the collaborative efforts of our contributors, we successfully obtained 12,248 images for AI generated scenarios, with approximately 1,000 images for each class. In summary, XIMAGENET-12 includes 12 classes across six scenarios, encompassing a total of 212,787 images. We have also secured all intellectual property rights for these synthesized images.

## 3 EXPERIMENT

### 3.1 Settings

We conducted benchmark tests on various SOTA object detection, classification, and segmentation models using XImageNet12 dataset. This encompassed a wide range of models from TensorFlow Keras, including the ResNet (11) series, MobileNet (29), EfficientNet (32) series, InceptionNet, DenseNet (15), and PyTorch framework models, including Transformer-based models like ViT (8). In addition, we included nine models from the MMDetection framework (1), such as Deeplabv3plus R50-D8 (3) and Upernet models (36) with ViT-B16 and LN-MLN transformers.

To ensure the validity of our results, we meticulously reproduced all model accuracy claims as documented in official Keras library documentation (16) (see Tab. 3 and Tab. 2). To mitigate random bias, we employed 12-fold cross-validation and reproduced the results over these folds.

We primarily conducted our experiments using TensorFlow and PyTorch, adhering to the standard hyperparameters provided in the official repositories. To quantitatively evaluate model robustness, we performed three distinct experiments: '**extra-scenarios**' (referred to as EX1), '**inter-scenarios**' (referred to as EX2), and '**AI-generated image segmentation**' (abbreviated as EX3 for simplicity on later sections of our paper). Each experiment involved multi-person verification to ensure reproducible results.

For **EX1, namely extra-scenarios** experiments, we used trained best weights from ImageNet original image and to test on our scenarios image to obtain the accuracy sensitivity for each scenario, because those scenarios were never learned by those visual models. For Tensorflow-based training, we adopted industry-recognized models such as ResNet-50 (11) and employed the SGD optimizer with (100 epochs) schedules, using a global batch size of 16 on a single T4 GPU, without data augmentation and input images had a resolution of 224×224 pixels. Transformer-based models, including ViT (8) models, utilized the Adam optimize under 0.0001 learning rate, with 200 epochs schedules and a global batch size of 16 distributed across 4 GPUs. Input images had a resolution of 224×224 pixels, with other settings mirroring those used in PyTorch-based experiments.

For **EX2, namely inter-scenarios**, we utilized representative models, including ResNet101 (11), MobileNetV3 (29), DenseNet121 (15) etc, alongside Transformer-based models like ViT (8). Our goal was to assess model robustness explicitly for each subclass objects, such as instruments and animals, within the same scenarios. We conducted all 12-fold classifications within 6 scenarios. We also incorporated early stopping and multi-scale training, using standardized resizing input images to dimensions between 224 and 224 pixels. For Transformer-based models, settings remained consistent within inter-scenario experiments.

For **EX3, namely segmentation** experiments, we focused on evaluating the performance of top-performing models, such as Swin-Transformer (22), ViT series (8), PsPNet (43), FPN (21), UperNet (37) and DeepLabV3 (2) with ResNet50 (11) as the backbone on synthetically AI-generated background images and their accuracy in predicting masks for problematic pixels along edges and boundaries. Training occurred over 40,000 iterations, with a global batch size of 4 distributed across 2 GPUs. Train image comes from original ImageNet with crop size of 256×256 pixels, 2582 images, and testing on our XimageNet-12 background scenarios images, including 2,463 AI-generated background images.

All testing was performed using original resolution input images, without any augmentation or learned models applied. Later we performed all **Multiple linear regression** on collect test accuracy results on each scenarios or sub classes to quantitatively prove the hypothesise we made.

Table 1: Performance Evaluation of Multiple Linear Regression in Experiment 1: Modeling Scenarios. With ResNet50 model and original image as baseline intercept.

| Variable | Estimate | Standard error | 95% CI (asymptotic) | P value | P value summary |
|---|---|---|---|---|---|
| Intercept | 0.8986 | 0.02238 | 0.8547 to 0.9426 | $< 0.0001$ | **** |
| Model Name[EfficientNetB0 (32) Best Weight] | -0.03444 | 0.01445 | -0.06282 to -0.006065 | 0.0175 | * |
| Model Name[EfficientNetB3 (32) Best Weight] | -0.04111 | 0.01445 | -0.06949 to -0.01273 | 0.0046 | ** |
| Model Name[DenseNet121 (15) Best Weight] | 0.01556 | 0.01445 | -0.01282 to 0.04394 | 0.2821 | ns |
| Model Name[MobileNetV2 (29) Best Weight] | 0.005556 | 0.01445 | -0.02282 to 0.03394 | 0.7007 | ns |
| Image Scenario[blur_background] | -0.0425 | 0.01938 | -0.08058 to -0.004424 | 0.0288 | * |
| Image Scenario[blur_object] | -0.07 | 0.01938 | -0.1081 to -0.03192 | 0.0003 | *** |
| Image Scenario[image_g] | -0.1257 | 0.01938 | -0.1637 to -0.08759 | $< 0.0001$ | **** |
| Image Scenario[image_b] | -0.0985 | 0.01938 | -0.1366 to -0.06042 | $< 0.0001$ | **** |
| Image Scenario[image_grey] | -0.06517 | 0.01938 | -0.1032 to -0.02709 | 0.0008 | *** |
| Image Scenario[image_r] | -0.087 | 0.01938 | -0.1251 to -0.04892 | $< 0.0001$ | **** |
| Image Scenario[Random Background with Real Environment] | -0.7078 | 0.01938 | -0.7459 to -0.6698 | $< 0.0001$ | **** |
| Image Scenario[Segmented_image] | -0.3012 | 0.01938 | -0.3392 to -0.2631 | $< 0.0001$ | **** |
| Image Class[1] | 0.134 | 0.02238 | 0.09003 to 0.1780 | $< 0.0001$ | **** |
| Image Class[2] | -0.04867 | 0.02238 | -0.09263 to -0.004701 | 0.0301 | * |
| Image Class[3] | 0.04 | 0.02238 | -0.003966 to 0.08397 | 0.0745 | ns |
| Image Class[4] | 0.1004 | 0.02238 | 0.05648 to 0.1444 | $< 0.0001$ | **** |
| Image Class[5] | 0.1333 | 0.02238 | 0.08937 to 0.1773 | $< 0.0001$ | **** |
| Image Class[6] | 0.07667 | 0.02238 | 0.03270 to 0.1206 | 0.0007 | *** |
| Image Class[7] | 0.01044 | 0.02238 | -0.03352 to 0.05441 | 0.6409 | ns |
| Image Class[8] | 0.09067 | 0.02238 | 0.04670 to 0.1346 | $< 0.0001$ | **** |
| Image Class[9] | 0.09933 | 0.02238 | 0.05537 to 0.1433 | $< 0.0001$ | **** |
| Image Class[10] | 0.1651 | 0.02238 | 0.1211 to 0.2091 | $< 0.0001$ | **** |
| Image Class[11] | -0.02244 | 0.02238 | -0.06641 to 0.02152 | 0.3164 | ns |

## 3.2 RESULTS AND FINDING: EX1

**Results on CNN-based methods.** We conducted multiple linear regression analyses (Fig. 10) to examine our hypotheses using the accuracy drop data (Tab. 1, 5). This regression model aimed to predict 'Classification Accuracy' based on three groups of predictor variables: 'Model Name,' 'Image Scenarios,' and 'Object Class.' Our overall model yielded statistical significance with $F(23, 516) = 99.40$ and $P < 0.0001$, indicating the relevance of at least one predictor variable in predicting the dependent variable. Specifically, 'Model Name' exhibited statistical significance with $F(4, 516) = 6.125$ and $P < 0.0001$. 'Image Scenarios' also showed significance with $F(8, 516) = 257.2$ and $P < 0.0001$. Lastly, 'Object Class' demonstrated statistical significance with $F(11, 516) = 18.54$ and $P < 0.0001$.

In our accuracy drop experiments, our experiments revealed a noticeable decline in accuracy, particularly when considering the accuracy density map Fig. 5 and variance Fig. 4. Notably, we observed that the presence of 'Random Background with Real Environment' had the most substantial adverse impact on accuracy, resulting in a significant decrease of 0.7078. Furthermore, the 'Segmented Image' scenario also exhibited a significant negative influence, leading to a decrease in accuracy by 0.3012 with a high level of statistical significance ($P < 0.0001$). This observation reinforces the idea that models trained exclusively on back-

Table 2: EX1 and EX2 performance Comparison of SOTA Models on Different Test/Train Datasets

| Pretrained Dataset | Model Name | Parameters (M) | Test Dataset (Top-1 Acc.) | | | | | | | |
|---|---|---|---|---|---|---|---|---|---|---|
| | | | Blur_bg | Blur_obj | Color_g | Color_b | Color_grey | Color_r | Rand_bg | Seg_img |
| ImageNet (Original images) | ResNet50 (11) | 25.60 | 90.97% | 88.17% | 84.42% | 86.98% | 92.13% | 89.03% | 22.41% | 68.55% |
| | VGG-16 (31) | 138.4 | 89.92% | 89.91% | 78.64% | 70.46% | 81.48% | 80.68% | 24.58% | 49.62% |
| | MobileNetV2 (29) | 3.5 | 92.34% | 88.52% | 85.73% | 88.67% | 88.81% | 89.33% | 27.14% | 66.43% |
| | EfficientNetB0 (32) | 5.3 | 91.44% | 90.86% | 78.10% | 82.45% | 86.44% | 83.65% | 25.29% | 53.56% |
| | EfficientNetB3 (32) | 12.3 | 86.80% | 84.53% | 77.99% | 81.22% | 83.00% | 83.85% | 22.06% | 69.67% |
| | DenseNet121 (15) | 8.1 | 93.77% | 88.92% | 87.39% | 87.33% | 93.23% | 88.21% | 26.41% | 69.67% |
| | ViT (8) | 86.6 | 88.44% | 90.77% | 65.87% | 62.82% | 70.69% | 66.53% | 17.21% | 49.01% |
| XImageNet-12 (*Corresponding Scenarios) | ResNet50 (11) | 25.60 | 83.52% | 80.24% | 83.61% | 84.45% | 84.71% | 80.40% | 53.91% | 85.76% |
| | VGG-16 (31) | 138.4 | 74.85% | 71.54% | 74.18% | 76.26% | 77.58% | 69.91% | 70.25% | 73.27% |
| | AlexNet (17) | 61.1 | 81.60% | 79.95% | 81.96% | 81.89% | 81.31% | 78.07% | 46.29% | 82.00% |
| | MobileNetV3 (13) | 3.50 | 67.36% | 67.88% | 72.04% | 74.25% | 74.25% | 64.79% | 43.33% | 78.85% |
| | DenseNet121 (15) | 8,1 | 90.79% | 86.57% | 88.92% | 89.96% | 90.44% | 87.37% | 69.58% | 91.60% |
| | ViT (8) | 86.56 | 71.51% | 70.21% | 74.77% | 75.96% | 75.80% | 71.14% | 38.01% | 78.69% |

grounds can significantly affect accuracy, highlighting the importance of background information in content reasoning. Our findings also support the hypothesis that more accurate models tend to be less reliant on background information. This is evident from the fact that the model with the highest top-1 accuracy during training exhibited less sensitivity to background scenarios (35).

Additionally, when considering specific model architectures, 'DenseNet121 (15) Best Weight' demonstrated a small increase in accuracy by 0.015. Similarly, 'MobileNetV2 (29) Best Weight' led to an accuracy increase of 0.005, which was less pronounced.

It is important to note that alterations to background color or image blurring had a negative impact on accuracy. However, this impact was less severe than the effect of introducing random backgrounds or complete background segmentation. Specifically, each of these modifications resulted in a noticeable difference of 0.1 to 0.2 percent, which remains statistically significant. Additionally, we found that 'Object Class 10' and 'Object Class 5' had the most positive impact on accuracy, leading to increases of 0.1651 and 0.1333, respectively. These effects were highly significant when compared to classes 6 and 7. This observation underscores the sensitivity of our explainable model to specific object classes, suggesting that the inherent attributes of objects play a significant role in content reasoning (see Tab. 2).

**Results on Transformer-based methods.** We conducted an evaluation of the Vision Transformer (ViT) model (8) using the PyTorch framework on the XimageNet-12 dataset. Our results, presented in Tab. 2, showcase the model's performance when pretrained on original images and tested on scenarios involving blurred backgrounds, with a mean Average Precision (mAP) of 88.43 for Blur Background and 90.76 for Blur Object images. Notably, ViT (8) demonstrates a substantial improvement in mAP, with a 26.4 increase when compared to Color Scenarios, for example, Image Green Channel (mAP 62.81)(see Tab. 2).

However, it's important to note that Transformer-based models, including ViT (8), have limitations in terms of speed and resource requirements. When processing images of the same resolution and batch size see Fig. 3 and Fig. 8, these models demand nearly twice the training time and GPU resources compared to other architectures. Consequently, there is a pressing need to develop Vision Transformers that are optimized for high-resolution images and industrial scenarios, striking a balance between accuracy and latency.

## 3.3 Results And Finding: EX2

We used mainstream classification models ResNet (11), VGG (31), AlexNet (17), MobileNet (29), and Vit (8) to train and test the within our scenarios. As shown in Fig.2, we can notice that after learning scenarios through fine-tuning, we observed a significant improvement in classification accuracy. Different models exhibit varying degrees of responsiveness to image transformations, with ViT (8) and DenseNet121 (15) demonstrating better robustness. An intriguing observation is that employing reasonable image segmentation

masks does not significantly hinder image classification results; in fact, it might even enhance classification accuracy. This finding aligns with some prior research, such as (19), which noted that models trained and tested with well-segmented foregrounds tend to exhibit improved performance. Therefore, we propose that the assertion in the work of (35), regarding a decrease in classification accuracy due to segmentation may be contingent on the quality of the segmentation rather than an inherent negative impact of segmentation on classification performance, see Fig.6 .

### 3.4 RESULTS AND FINDING: EX3

Here, we conducted a regression analysis to assess segmentation accuracy. Our reference model was Deeplabv3plus (2) R50-D8, and the fundamental images served as the baseline. We ran seven segmentation experiments, including state-of-the-art models like Deeplabv3plus (2) R50-D8 and Upernet (37) models with ViT-B16 (8) and LN-MLN (1) transformers. For detailed segmentation accuracy data, refer to the Tab.6. Our regression analysis demonstrated a significant relationship between variables and segmentation accuracy, with highly significant F-statistics $P < 0.0001$. Introducing a generated background scenario led to a 14.08% accuracy decrease compared to the original scenario. Specifically, the DPT ViT (8) B16 model had a 19.99% accuracy decrease, the Upernet Swin (22), (37) model showed a 22.11% decrease, the Upernet ViT B16 LN MLN (37), (8), (1) model had a 16.95% decrease, the FPN (21) R50 model exhibited a 20.81% decrease compared to the reference model.

## 4 OUR ROBUSTNESS SCORES FRAMEWORK

In general, we face the challenge of lacking a standard reference for evaluating the robustness scores of visual models. Currently, most methods rely on testing these models on well-known benchmarks, often leading researchers to test on more and more datasets (23), (39). However, testing on additional datasets doesn't necessarily make a model more robust. Importantly, these methods can assess a model's generalization performance but cannot verify its performance in different scenarios, such as changes in lighting, blurring, or background noise.

Moreover, while precision and recall have clear mathematical formalizations, robustness lacks a similar formalization. Therefore, inspired by the mathematical definitions of variance and covariance, we have developed robustness scores based on our explainable benchmarks, as follows:

$$(\sigma_e)^2 = \frac{\sum_{i=1}^{n}(C(i) - \mu)^2}{n}$$

, where $\mu$ is Expect Best Weight Acc. namely, best test Accuracy on the original image with pre-trained under original scenarios, with respect to $n$ ( scenarios number, such as blur, color...).

$$(\sigma_i)^2 = \frac{\sum_{k=1}^{n}(C'(k) - \mu)^2}{n}$$

where $\{0, 1, \ldots, n\}$ represents the set of all target object subclasses between $0$ and $n$, and $\mu$ denotes the Expected Best Weight Accuracy, i.e., the best test accuracy on the original image under original scenarios. In the end, Robustness can be categorized into two key aspects: Internal scenario object class robustness, which pertains to sensitivity to different classes of objects, such as animals and humans, etc. External scenario robustness signifies the ability to maintain stability amidst changing background conditions for the same objects:

$$S_{\text{robust}} = 1 - (\sigma_i + \sigma_e)$$

In addition to its statistical usage, "variance" is a term employed in the realm of for example finance. In finance, stock analysts and financial advisors employ variance to gauge a stock's volatility or level of uncertainty. We also made significant comparisons for three types of classification models using our robustness score in Tab.3.

## 5 ABLATION STUDY: COMPARING ON EXISTING BENCHMARK

There has been prior work on mitigating contextual bias in image classification (9), (12), the influence of background signals on various datasets, and techniques like foreground segmentation that we leverage. In the realm of image backgrounds, prior studies have uncovered the predictive nature of background correlations (33). These investigations have also shed light on the varying degrees to which backgrounds can impact model decisions (45), (27).

Our research shares commonalities with the work of Zhu, Xie, and Yuille (45), who delved into ImageNet classification and segmentation. However, we've taken several significant steps forward: (a) we provide a comprehensive toolkit for measuring model robustness across six scenarios, (b) we've enhanced semantic labeling quality and introduced cross-validation to eliminate noise and inaccuracies, (c) we investigate model robustness against adversarial backgrounds, and (d) we consider a wider array of contemporary image classifiers. In essence, our study contributes to a deeper comprehension of content understanding by incorporating modern models and examining diverse factors influencing their predictions.

## 6 DISCUSSION AND CONCLUSION

In conclusion, the motivation behind the creation of this dataset stems from our experiences with anomaly detection models deployed in industry projects. Through these deployments, we have encountered various unexpected scenarios where model performance was challenged. Our primary objective in developing this dataset was to bridge the gap between academia and industry. By offering an explainable AI dataset XIMAGENET-12 where the target object remains consistent while the background introduces noise or undergoes alterations, and an easily comprehensible robustness scoring mechanism, we enable researchers and practitioners to evaluate and enhance the true robustness of their models in context. In essence, this dataset serves as a critical tool to help the industry and academia collaborate effectively, ensuring that computer vision models are not only accurate in controlled settings but also capable of withstanding the complexities and uncertainties.

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

# 7 APPENDIX

You may include other additional sections here.

Table 3: Variance of Model Accuracy Performance and Robustness Scores

| Model Acc. Drop Volatility | Scenarios | | | | | | | | Variance | Robustness Score (Our* 0 - 1) | Offical Top-1 Acc. (On ImageNet) |
| --- | --- | --- | --- | --- | --- | --- | --- | --- | --- | --- | --- |
| | Blur_background | Blur_object | Image_g | Image_b | Image_grey | Image_r | Random_background | Segmented_image | | | |
| ResNet50 (11): external | 0,18% | 0,50% | 1,17% | 0,68% | 0,10% | 0,38% | 53,04% | 7,12% | 0,0902 | 0,8985 | 74,90% |
| ResNet50 (11): internal | 0,10% | 0,00% | 0,11% | 0,17% | 0,20% | 0,00% | 6,96% | 0,30% | 0,0112 | | |
| DenseNet121 (15):external | 0,13% | 0,72% | 1,00% | 1,01% | 0,17% | 0,84% | 50,39% | 7,69% | 0,0885 | 0,9062 | 75,00% |
| DenseNet121 (15):internal | 0,21% | 0,00% | 0,07% | 0,14% | 0,18% | 0,01% | 2,77% | 0,29% | 0,0052 | | |
| VGG-16 (31):external | 0,15% | 0,15% | 2,30% | 5,45% | 1,52% | 1,72% | 47,93% | 19,53% | 0,1125 | 0,8845 | 71,30% |
| VGG-16 (31):internal | 0,33% | 0,06% | 0,26% | 0,51% | 0,72% | 0,01% | 0,01% | 0,17% | 0,0029 | | |
| ViT (8):external | 0,25% | 0,07% | 7,60% | 9,38% | 5,18% | 7,24% | 58,11% | 19,74% | 0,1536 | 0,8196 | 81,07% |
| ViT (8):internal | 0,51% | 0,72% | 0,15% | 0,07% | 0,08% | 0,57% | 16,54% | 0,00% | 0,0266 | | |

Table 4: Analysis of Variance and Parameter Estimates

| Analysis of Variance | EX1 | | | | EX3 | | | |
| --- | --- | --- | --- | --- | --- | --- | --- | --- |
| | SS | DF | MS | F (DFn, DFd) / P value | SS | DF | MS | F (DFn, DFd) / P value |
| Regression | 25.76 | 23 | 1.12 | $F(23, 516) = 99.40 / < 0.0001$ | 17.47 | 25 | 0.6989 | $F(25, 730) = 41.95 / < 0.0001$ |
| Model Name | 0.2761 | 4 | 0.06903 | $F(4, 516) = 6.125 / < 0.0001$ | 5.418 | 6 | 0.9029 | $F(6, 730) = 54.20 / < 0.0001$ |
| Image Scenario | 23.19 | 8 | 2.899 | $F(8, 516) = 257.2 / < 0.0001$ | 2.4 | 8 | 0.3 | $F(8, 730) = 18.01 / < 0.0001$ |
| Image Class | 2.298 | 11 | 0.2089 | $F(11, 516) = 18.54 / < 0.0001$ | 9.655 | 11 | 0.8777 | $F(11, 730) = 52.68 / < 0.0001$ |
| Residual | 5.815 | 516 | 0.01127 | | 12.16 | 730 | 0.01666 | |
| Total | 31.58 | 539 | | | 29.63 | 755 | | |

Table 5: Classification Accuracy Density Map of DenseNet121 (15), pretrained on ImageNet, Test on XImageNet12. This table contains the content for the DenseNet121 (15) model results with the density map-like color scale for accuracy.

| Model Name | Background Scenarios | Class Name | | | | | | | | | | | |
| --- | --- | --- | --- | --- | --- | --- | --- | --- | --- | --- | --- | --- | --- |
| | | 0 | 1 | 2 | 3 | 4 | 5 | 6 | 7 | 8 | 9 | 10 | 11 |
| DenseNet121 (15) | blur_background | 0.95 | 0.97 | 0.89 | 0.93 | 0.93 | 0.96 | 0.89 | 0.91 | 0.98 | 0.93 | 0.99 | 0.92 |
| | blur_object | 0.74 | 0.91 | 0.72 | 0.91 | 0.93 | 0.94 | 0.73 | 0.95 | 0.94 | 0.96 | 0.98 | 0.97 |
| | image_g | 0.81 | 0.93 | 0.75 | 0.68 | 0.89 | 0.94 | 0.9 | 0.92 | 0.9 | 0.98 | 0.98 | 0.8 |
| | image_b | 0.71 | 0.77 | 0.75 | 0.82 | 0.96 | 0.97 | 0.93 | 0.89 | 0.96 | 0.89 | 0.96 | 0.87 |
| | image_grey | 0.94 | 0.93 | 0.83 | 0.81 | 0.98 | 0.97 | 0.95 | 0.98 | 0.9 | 0.98 | 0.98 | 0.95 |
| | image_r | 0.87 | 0.87 | 0.82 | 0.92 | 0.97 | 0.96 | 0.94 | 0.85 | 0.92 | 0.89 | 0.97 | 0.57 |
| | Random Background with Real Environment | 0.3 | 0.32 | 0.2 | 0.39 | 0.16 | 0.32 | 0.09 | 0.07 | 0.14 | 0.46 | 0.63 | 0.07 |
| | Segmented_image | 0.63 | 0.85 | 0.67 | 0.51 | 0.91 | 0.83 | 0.73 | 0.62 | 0.75 | 0.69 | 0.95 | 0.16 |
| | Original | 0.96 | 0.97 | 0.93 | 0.97 | 0.98 | 0.98 | 0.97 | 0.99 | 0.99 | 0.98 | 0.98 | 0.99 |

Table 6: Multiple linear regression performance of Experiment 3 on Model about Segmentation Tasks

| Variable | Estimate | Standard error | 95% CI (asymptotic) | P value | P value summary |
|----------|----------|----------------|---------------------|---------|-----------------|
| Intercept | 0.6672 | 0.02394 | 0.6202 to 0.7142 | < 0.0001 | **** |
| Model Name[dpt_vit-b16 (8)] | -0.1999 | 0.01756 | -0.2344 to -0.1654 | < 0.0001 | **** |
| Model Name[upernet_swin (37)] | -0.2211 | 0.01756 | -0.2556 to -0.1866 | < 0.0001 | **** |
| Model Name[upernet_vit-b16 _ln_mln(37)] | -0.1695 | 0.01756 | -0.2040 to -0.1351 | < 0.0001 | **** |
| Model Name[pspnet_r50-d8 (43)] | -0.045 | 0.01756 | -0.07948 to -0.01052 | 0.0106 | * |
| Model Name[fpn_r50 (21)] | -0.2081 | 0.01756 | -0.2426 to -0.1737 | < 0.0001 | **** |
| Model Name[upernet_r50 (37)] | -0.05796 | 0.01756 | -0.09245 to -0.02348 | 0.001 | ** |
| Image Scenario[blur_background] | 0.01833 | 0.01992 | -0.02077 to 0.05743 | 0.3576 | ns |
| Image Scenario[blur_object] | -0.1571 | 0.01992 | -0.1962 to -0.1180 | < 0.0001 | **** |
| Image Scenario[image_g] | -0.07131 | 0.01992 | -0.1104 to -0.03221 | 0.0004 | *** |
| Image Scenario[image_b] | -0.03952 | 0.01992 | -0.07862 to -0.0004243 | 0.0476 | * |
| Image Scenario[image_grey] | -0.01929 | 0.01992 | -0.05839 to 0.01981 | 0.3332 | ns |
| Image Scenario[image_r] | -0.07702 | 0.01992 | -0.1161 to -0.03792 | 0.0001 | *** |
| Image Scenario[segmented_image] | -0.08143 | 0.01992 | -0.1205 to -0.04233 | < 0.0001 | **** |
| Image Scenario[generated_background] | -0.1408 | 0.01992 | -0.1799 to -0.1017 | < 0.0001 | **** |
| Image Class[1] | 0.07619 | 0.023 | 0.03104 to 0.1213 | 0.001 | *** |
| Image Class[2] | -0.06508 | 0.023 | -0.1102 to -0.01993 | 0.0048 | ** |
| Image Class[3] | 0.05222 | 0.023 | 0.007074 to 0.09737 | 0.0234 | * |
| Image Class[4] | 0.08127 | 0.023 | 0.03612 to 0.1264 | 0.0004 | *** |
| Image Class[5] | 0.2713 | 0.023 | 0.2261 to 0.3164 | < 0.0001 | **** |
| Image Class[6] | 0.3021 | 0.023 | 0.2569 to 0.3472 | < 0.0001 | **** |
| Image Class[7] | 0.1641 | 0.023 | 0.1190 to 0.2093 | 7.137 | **** |
| Image Class[8] | 0.1548 | 0.023 | 0.1096 to 0.1999 | 6.73 | **** |
| Image Class[9] | 0.2216 | 0.023 | 0.1764 to 0.2667 | 9.635 | **** |
| Image Class[10] | 0.2689 | 0.023 | 0.2237 to 0.3140 | 11.69 | **** |
| Image Class[11] | 0.04079 | 0.023 | -0.004355 to 0.08594 | 1.774 | ns |

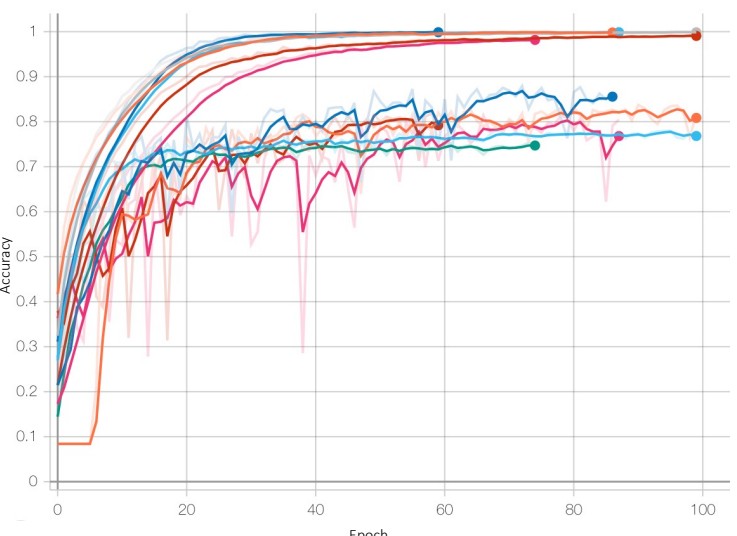

Training accuracy for SOTA models for Experiment 1

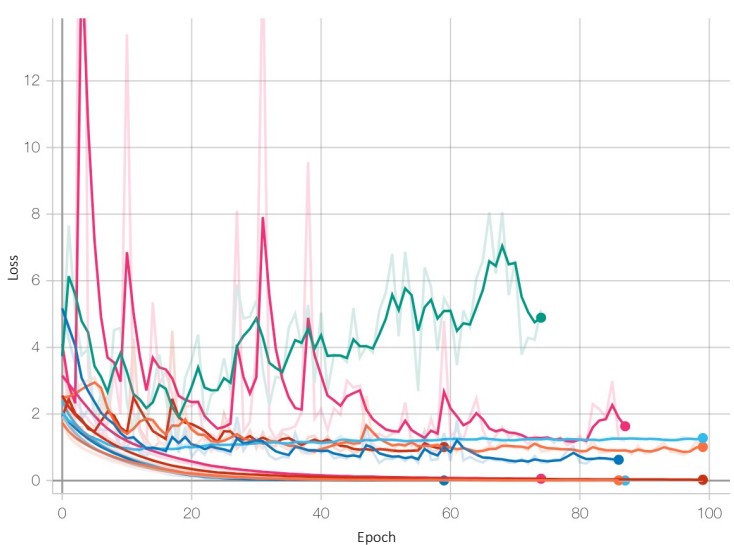

Training losses for SOTA models for Experiment 1

| Name | Smoothed | Value | Step | Time | Relative |
| --- | --- | --- | --- | --- | --- |
| DenseNet121/train | 0.9984 | 0.9994 | 72 | Thu Sep 14, 23:33:58 | 3h 8m 27s |
| DenseNet121/validation | 0.8639 | 0.8793 | 72 | Thu Sep 14, 23:33:59 | 3h 8m 27s |
| EfficientNetB0/train | 0.9836 | 0.9833 | 72 | Wed Sep 13, 19:57:54 | 1h 4m 23s |
| EfficientNetB0/validation | 0.7709 | 0.7706 | 72 | Wed Sep 13, 19:57:54 | 1h 4m 23s |
| EfficientNetB3/train | 0.9804 | 0.9814 | 72 | Wed Sep 13, 22:42:08 | 1h 47m 29s |
| EfficientNetB3/validation | 0.7429 | 0.7466 | 72 | Wed Sep 13, 22:42:08 | 1h 47m 29s |
| MobileNetV2/train | 0.9958 | 0.9965 | 72 | Thu Sep 14, 18:06:33 | 1h 12m 15s |
| MobileNetV2/validation | 0.8015 | 0.8188 | 72 | Thu Sep 14, 18:06:33 | 1h 12m 15s |
| ResNet101V2/train | 0.9986 | 0.9991 | 59 | Fri Sep 15, 04:41:36 | 3h 47m 54s |
| ResNet101V2/validation | 0.7922 | 0.7822 | 59 | Fri Sep 15, 04:41:36 | 3h 47m 54s |
| ResNet50/train | 0.998 | 0.9982 | 72 | Thu Sep 14, 15:36:17 | 2h 53m 49s |
| ResNet50/validation | 0.7903 | 0.8058 | 72 | Thu Sep 14, 15:36:17 | 2h 53m 49s |

Training details for SOTA models

Figure 3: Training details and loss plot for Experiment 1

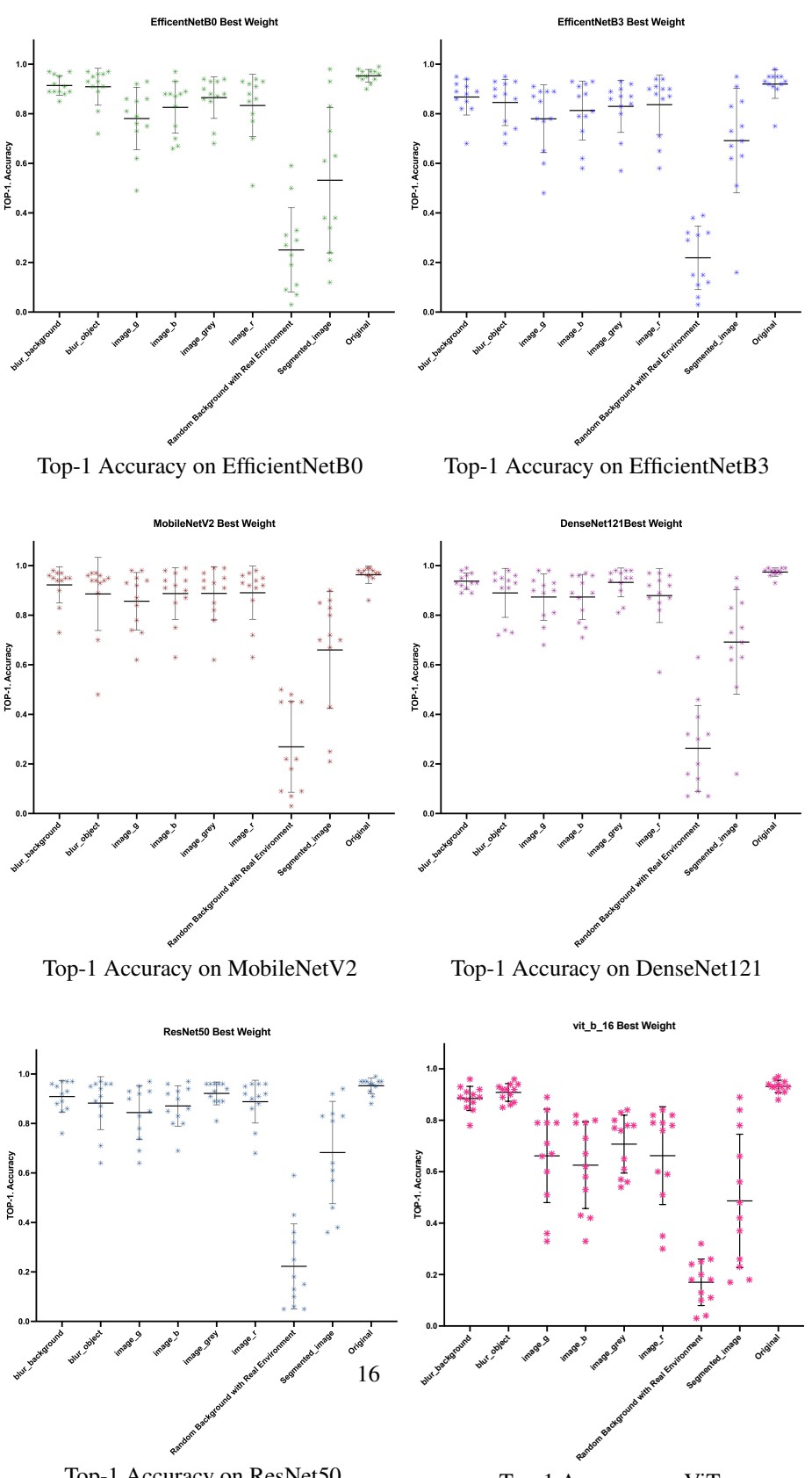

Top-1 Accuracy on EfficientNetB0 Top-1 Accuracy on EfficientNetB3

Top-1 Accuracy on MobileNetV2 Top-1 Accuracy on DenseNet121

16

Top-1 Accuracy on ResNet50 Top-1 Accuracy on ViT

Figure 4: TOP-1 Accuracy for SOTA models pre-trained on IMAGENET original images and tested on XIMAGENET-12

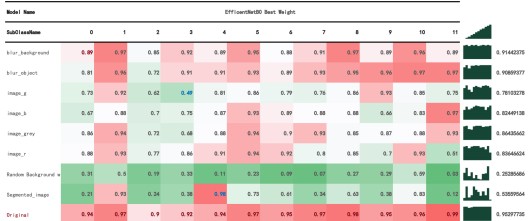

The EfficientNetB0 accuracy for each class

The EfficientNetB3 accuracy for each class

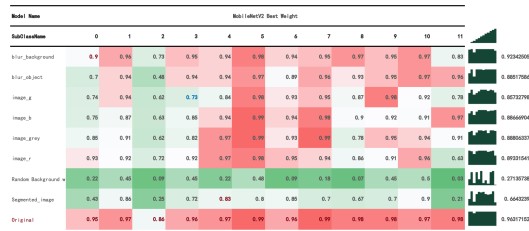

The MobileNetV2 accuracy for each class

The ViT accuracy for each class

The DenseNet121 accuracy for each class

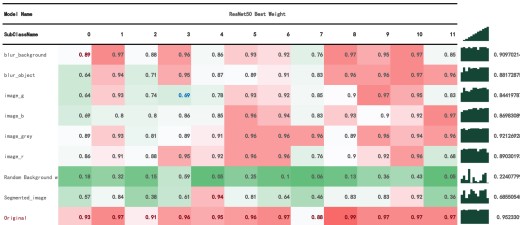

The ResNet50 accuracy for each class

Figure 5: The SOTA models accuracy density map for each class on Experiment 1

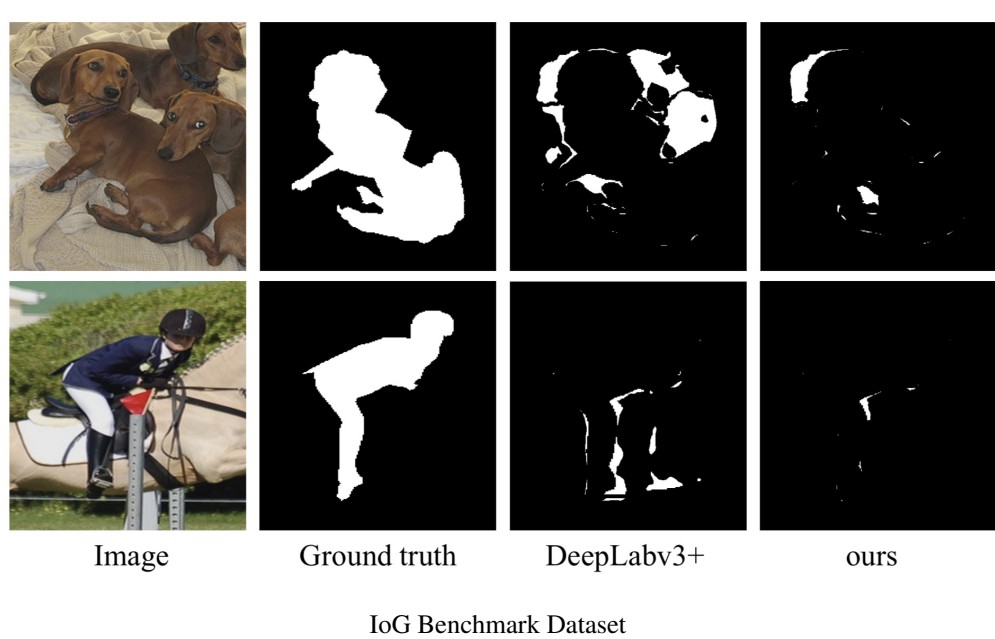

| Image | Ground truth | DeepLabv3+ | ours |

IoG Benchmark Dataset

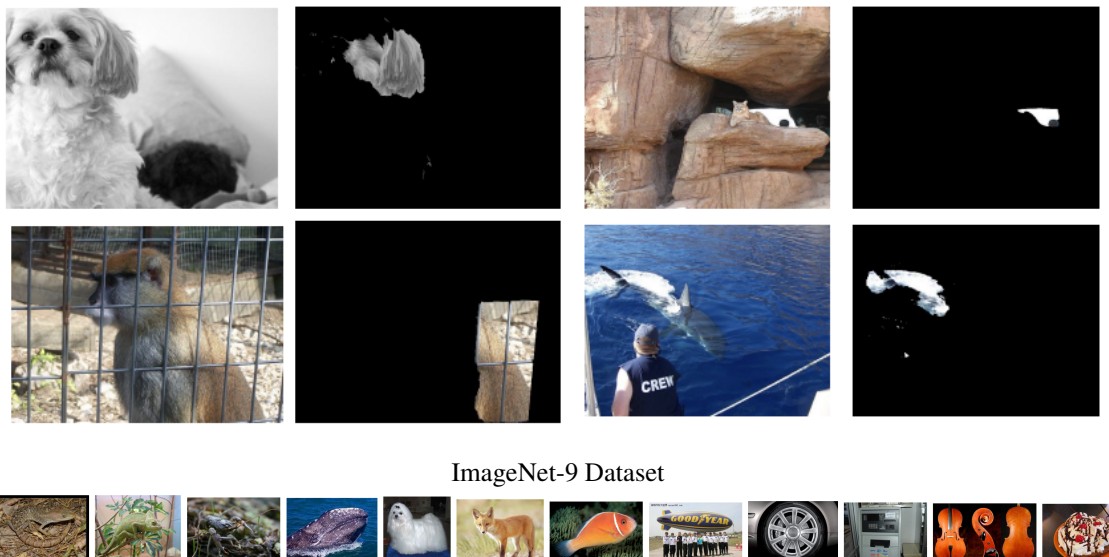

ImageNet-9 Dataset

Our XIMAGENET-12 mask samples

Figure 6: Additional Related Works and Explicit Comparisons

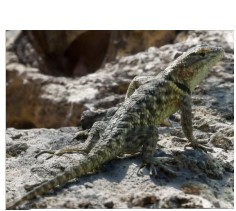 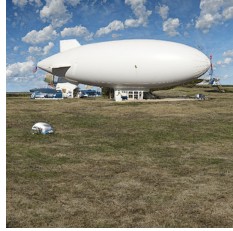 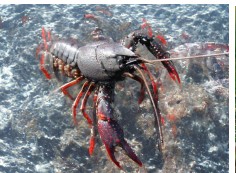 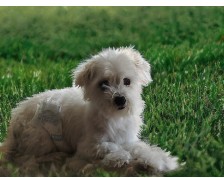

(a) **Prompt**: Generate high-definition pictures like those in the National Geographic magazine, keep the background unchanged.

(b) **Prompt**: Generate a realistic blue sky, and clouds background and please do not change the foreground airship object.

(c) **Prompt**: Generate high resolution images in sea water

(d) **Prompt**: Generate a picture with a foreground and the green grass in the background, similar to the official HD picture released by the state.

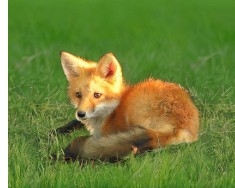 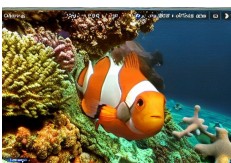 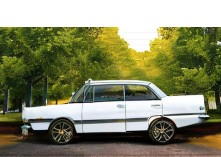 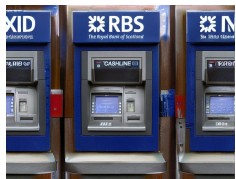

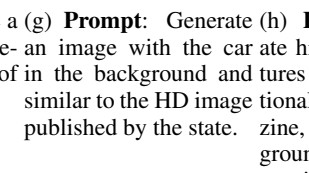

(e) **Prompt**: Generate high-resolution pictures like fox in the lawn, National Geographic, keep the background and foreground more simple and real.

(f) **Prompt**: Generate a simple image more realistic in the style of ocean magazine.

(g) **Prompt**: Generate an image with the car in the background and similar to the HD image published by the state.

(h) **Prompt**: Generate high-resolution pictures like those in National Financial Magazine, and keep the background and foreground consistent and the environment more real!

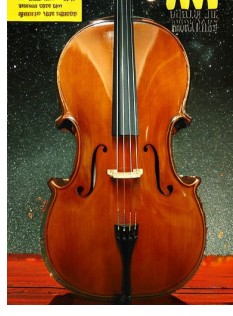 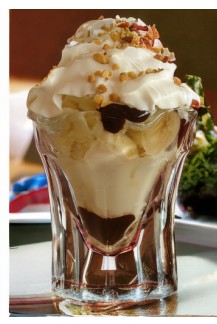 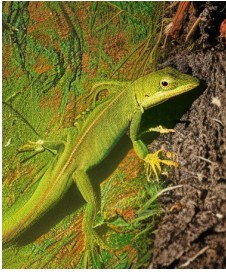 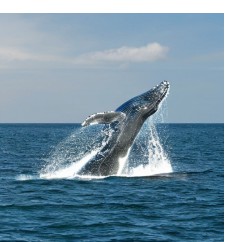

(i) **Prompt**: Please generate high-resolution pictures like those in the National Music Magazine and keep the background unchanged.

(j) **Prompt**: Generate high-resolution pictures in the style of those in National Food Magazine, and keep the background and foreground consistent and the environment more real!

(k) **Prompt**: Generate high-definition pictures like those in the National Geographic magazine, keep the background unchanged.

(l) **Prompt**: Generate high-resolution pictures, such as National Marine Magazine's oceans and whales, to keep the background real.

Figure 7: AI generated images with prompts

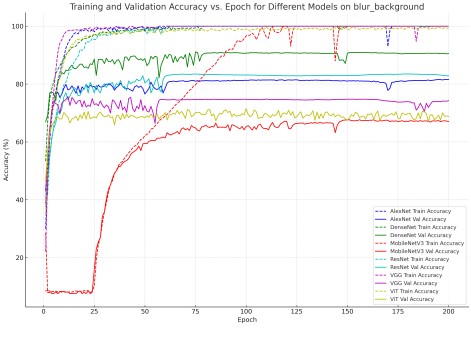

Top-1 training and validation accuracy on blurred background images

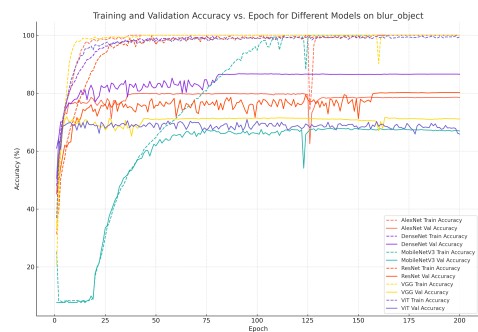

Top-1 training and validation accuracy on blurred object images

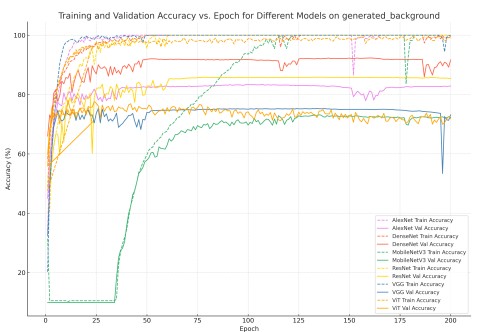

Top-1 training and validation accuracy on AI-generated background scenario

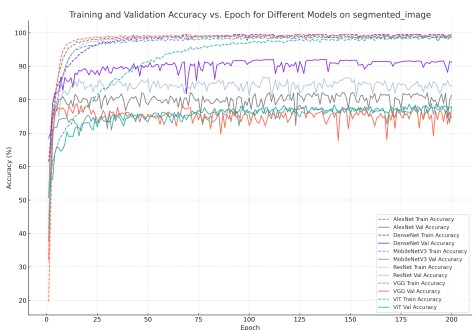

Top-1 training and validation accuracy on the segmented scenario

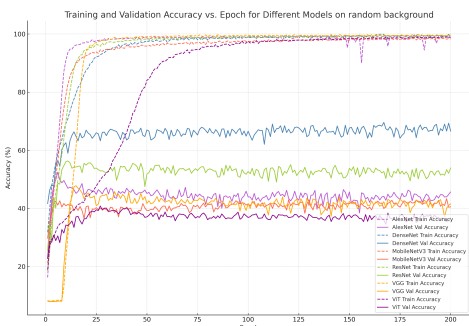

Top-1 training and validation accuracy on random background-generated scenario

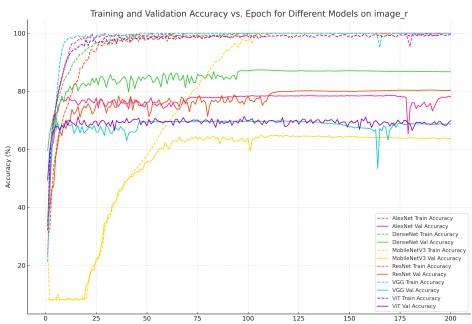

Top-1 training and validation accuracy on only one red channel generated scenario

Figure 8: Top-1 training and validation accuracy for SOTA models on Experiment 2

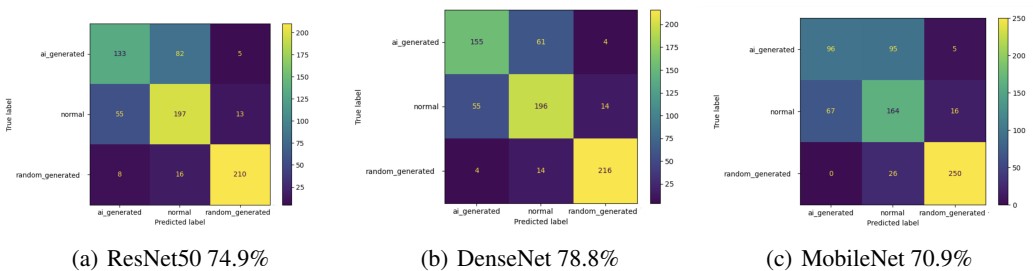

(a) ResNet50 74.9%       (b) DenseNet 78.8%       (c) MobileNet 70.9%

Figure 9: Model accuracy for classifying normal/AI-generated/random-generated images

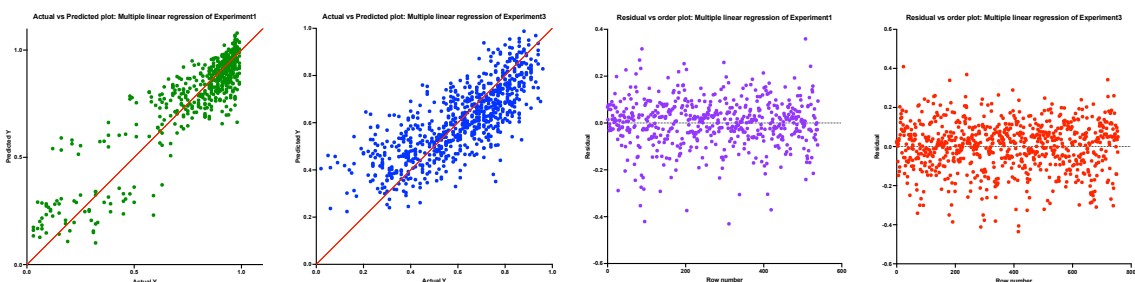

Figure 10: Multiple linear regression performance of Experiment 1 and Experiment 3

