# OpenReview forum: "XIMAGENET-12: An Explainable AI Benchmark Dataset for Model Robustness Evaluation"
_ICLR.cc/2024/Conference — ICLR 2024 Conference Withdrawn Submission_

### Official Review · Reviewer_Cwsn · 2023-10-30

**Soundness:** 1 poor
**Presentation:** 1 poor
**Contribution:** 2 fair
**Rating:** 3
**Confidence:** 5

**Summary:**

In this paper, a robustness dataset and metrics are proposed to bridge the gap between academia and industry. But the paper is not well organized. The illustration of the proposed metrics and experiment setting are not clear. Besides, the dataset should be evaluated with more state-of-the-art methods.

**Strengths:**

1.	An explainable AD dataset XIMAGENET-12 is generated to s to bridge the gap between academia and industry. The XIMAGENET-12 dataset possesses various annotation categories, relatively high image information quality, and public availability.

2.	A model robustness score formal schema is proposed to evaluate the model’s robustness.

**Weaknesses:**

1.	The paper is not well organized. The Section4 our robustness score framework should be in front of the Section3 Experiment.

2.	The dataset should be evaluated with the state-of-the-art backbone. The lasted model is ViT proposed in 2020 in your paper only in Ex3. The figures are to big and not well typesetted.

3.	The formulation of the proposed robustness score is confused. The difference of the first two equations are not illustrated, such as C and C’.

4.	Lack of illustration that how the dataset can bridge the gap between academia and industry

5.	Lack of result evaluation for Ex2 and Ex3. And in Ex2, it seems you should cite Tab.2 instead of Fig2 in Section3.3.

6.	The evaluation metrics are not explained in the experiments

7.	The evaluation of the proposed robustness score should be in the main body not in the appendix.

**Questions:**

1.	What’s the meaning of the evaluation metrics in Table1, Table2.

2.	In Results on Transformer-based methods. Blur Background has more influence than Blur Object images. What’s the reason?

3.	Many results in appendix seems to be redundant. They are not mentioned in the main body or the appendix, such as Fig3, Fig8

4.	Some typos:

a)	the first GenAI in contribution1 is needless.

b)	in Ex2, it seems you should cite Tab.2 instead of Fig2 in Section3.3.

---

### Official Review · Reviewer_eyGx · 2023-10-30

**Soundness:** 2 fair
**Presentation:** 2 fair
**Contribution:** 3 good
**Rating:** 5
**Confidence:** 3

**Summary:**

This paper introduces XIMAGENET-12, a new benchmark dataset for evaluating model robustness in explainable AI settings. The dataset contains over 200K images across 12 ImageNet categories with manual pixel-level annotations. It covers 6 scenarios like blurring, color changes, and generated backgrounds to simulate real-world variations. The authors propose a robustness score based on accuracy variance across scenarios and classes. Through experiments on classification, detection, and segmentation models, they find that introducing generated backgrounds causes significant performance drops. The key contributions are:
1. XIMAGENET-12, a large-scale robustness benchmark with pixel labels and simulated scenarios.
2. A robustness scoring method based on accuracy variance.
3. Analysis showing transformer and CNN models are sensitive to generated backgrounds.

**Strengths:**

1. The dataset provides plenty of semantic annotation, which is useful for robustness research on segmentation task.
2. The scenarios are diverse and simulate practically relevant issues like blurring.
3. The robustness score provides a simple metric for model evaluation and comparison.

**Weaknesses:**

1. More details could be provided on the data annotation process and quality control.
2. The robustness score could be evaluated more thoroughly as a metric.
3. The writing quality could be improved in some areas for clarity and concision.
4. Ablation studies compared to other datasets are limited.
5. This benchmark seems like a two-task benchmark, only for segmentation and classification.

**Questions:**

This article uses a large number of synthetic images, and there is a gap between natural images and synthetic images. Are the training and conclusions on synthetic images useful for real-world scenarios and nature images?

---

### Official Review · Reviewer_WJcw · 2023-11-02

**Soundness:** 1 poor
**Presentation:** 1 poor
**Contribution:** 1 poor
**Rating:** 3
**Confidence:** 5

**Summary:**

This paper proposed a dataset for measure the robustness of vision models, the dataset covers 12 categories from ImageNet, and with manual labeling of the segmentation mask, 6 scenarios with different background are introduce to test the robustness of vision models. Experiments show some what a performance degradation of state-of-the-art vision models on the proposed benchmark.

**Strengths:**

1. I think the P value in the evaluation is a good point.
2. Testing images with different background indeed is a challenging scenario for vision models.

**Weaknesses:**

1. The presentation of paper is not clear, the paper did not even give the name of the 12 selected categories.
2. The paper claims that the proposed dataset bridge the gap between academia and industry anomaly detection models, but from the current text, it is hard to see the relationship of the proposed benchmark to anomaly detection.
3. It seems that the images is taken from the ImageNet training set (since each categories have 1300 images which is the number of images in the training set for each class), how these training set images can be used for evaluating the performance of models that have already been trained on ImageNet?
4. It is very hard to parse the main results of the evaluation.

**Questions:**

1. Is that images from the ImageNet training set? If so, what is the rationale that these images can be used to test models that have already trained on it?
2. Please summarize the main results or conclusions of the evaluation.

---

### Author Response · Authors · 2023-11-18
**Thank you!**

We thank all the reviewers for their careful reading and detailed comments. These comments helped us a lot to improve our paper in writing. We decided to carefully revise our paper. We promise that you will see a much better version when we submit our paper next time!